# Exosomal RNAs: Novel Potential Biomarkers for Diseases—A Review

**DOI:** 10.3390/ijms23052461

**Published:** 2022-02-23

**Authors:** Jian Wang, Bing-Lin Yue, Yong-Zhen Huang, Xian-Yong Lan, Wu-Jun Liu, Hong Chen

**Affiliations:** 1Key Laboratory of Animal Genetics, Breeding and Reproduction of Shaanxi Province, College of Animal Science and Technology, Northwest A&F University, Yangling 712100, China; wangjsci@126.com (J.W.); hyzsci@nwafu.edu.cn (Y.-Z.H.); lanxianyong79@nwsuaf.edu.cn (X.-Y.L.); 2Key Laboratory of Qinghai-Tibetan Plateau Animal Genetic Resource Reservation and Utilization, Sichuan Province and Ministry of Education, Southwest Minzu University, Chengdu 610225, China; yuebinglin123@163.com; 3College of Animal Science, Xinjiang Agricultural University, Urumqi 830052, China

**Keywords:** exosome, biomarker, human disease, mRNA, miRNA, lncRNA, circRNA, isolation techniques

## Abstract

Exosomes are a subset of nano-sized extracellular vesicles originating from endosomes. Exosomes mediate cell-to-cell communication with their cargos, which includes mRNAs, miRNAs, lncRNAs, and circRNAs. Exosomal RNAs have cell specificity and reflect the conditions of their donor cells. Notably, their detection in biofluids can be used as a diagnostic marker for various diseases. Exosomal RNAs are ideal biomarkers because their surrounding membranes confer stability and they are detectable in almost all biofluids, which helps to reduce trauma and avoid invasive examinations. However, knowledge of exosomal biomarkers remains scarce. The present review summarizes the biogenesis, secretion, and uptake of exosomes, the current researches exploring exosomal mRNAs, miRNAs, lncRNAs, and circRNAs as potential biomarkers for the diagnosis of human diseases, as well as recent techniques of exosome isolation.

## 1. Introduction

Disease diagnosis is a key step in clinical practice. The detection of soluble biomarkers from biofluids became a critical method in the early diagnosis of diseases. Exosomes received increased attention due to their wide distribution in body fluids and their ability to reflect physiological and pathological conditions.

Exosomes are a type of lipid membrane-bound extracellular vesicle with an average size of 100 nm (ranging from ~40 to 160 nm) [1]. Most cell types, including mesenchymal stem cells, endothelial cells, myoblasts, and adipocytes, can release exosomes of different sizes, compositions, and functions. More importantly, exosomes are widely present in almost all biofluids, including cell supernatant, blood, plasma, saliva, urine, serum, and breast milk [2,3,4,5,6]. Exosomal contents, including RNAs, DNAs, proteins, and lipids, can participate in physiological processes such as intercellular communication and material transport [7]. There is particularly strong evidence of exosomal RNAs regulating gene expression and function in recipient cells. Exosomal RNAs can affect normal physiological metabolic activities and participate in the development of various diseases, including tumor growth, neurodegenerative disease, and metabolic syndrome [7,8,9,10]. Exosomal RNAs are a promising source of diagnostic biomarkers for human diseases [11,12]. In this review, we focus on recent studies exploring exosomal mRNAs and ncRNAs (miRNAs, lncRNAs, and circRNAs) as biomarkers for human diseases.

## 2. Biogenesis of Exosomes

Exosomes are extracellular vesicles of endosomal origin. First, the plasma membrane invaginates to form an early endosome. This endocytosis process can encapsulate extracellular soluble proteins, and the early endosome membrane can contain cell-surface proteins [13]. Then, the early endosomes give rise to late endosomes, invaginate, and form intraluminal vesicles (ILVs) [13]. Here, the late-stage endosomal structures containing dozens of ILVs are typically known as multivesicular bodies (MVBs), which are partly delivered to lysosomes for degradation or fusion with the plasma membrane to release the contained ILVs as exosomes (Figure 1). Specifically, the formation of the endosomal sorting complexes required for transport (ESCRT) is essential for both the synthesis and secretion of exosomes. The ESCRTs consist of four complexes (ESCRT-0, I, II, and III) and related proteins (VPS4, TSG101, and ALIX) [14,15]. ESCRT-0 classifies ubiquitin cargo proteins into lipid domains, whereas ESCRT-I and ESCRT-II are responsible for deforming the membrane to format a stable membrane neck [14]. ESCRT-III participates in membrane deformation and fission, such as promoting ILV budding [16,17]. The recruitment of the VPS4 complex into ESCRT-III results in vesicle neck dissection and the dissociation and recycling of the ESCRT-III complex [18]. TSG101 is connected to the release of exosomes, and the activation of ALIX protein could recruit ESCRT-III proteins to endosomes [16,19]. Additionally, numerous studies showed that exosome synthesis and cargo loading involve an ESCRT-independent pathway using lipids and associated proteins [20,21,22] (Figure 1). 

RNA loading into exosomes appears to be lipid-mediated and depends on specific self-organizing lipids and cargo domains. Specific nucleotide sequences have enhanced affinity for the phospholipid bilayer, including lipid rafts, hydrophobic modifications, or sphingosine [23]. Lipid rafts are highly enriched in cholesterol, sphingolipids, and glycosylphosphatidylinositol-anchored proteins, and their binding to proteins or other molecules might promote their secretion through exosomes [24]. Additionally, the presence of ceramide, lysophosphatidic, and glycosphingolipid molecules on the limiting membrane causes a spontaneous budding process that results in the formation of ILVs [25]. Meanwhile, ceramide may be transformed to sphingosine and sphingosine-1-phosphate (S1P) by the enzymes ceramide kinase and ceramidase, while the subsequent activation of S1P receptors on the limiting membrane facilitates the mediation of tetraspanin sorting into ILVs [26,27]. The tetraspanin superfamily consists of cell surface-associated membrane proteins characterized by transmembrane domains and organizes the membrane microdomains known as tetraspanin-enriched microdomains, which contain abundant transmembrane and cytoplasmic signal proteins [28]. Notably, although it was reported that the absence of ESCRT machinery did not prevent the production of MVBs in mammalian cells, it may bring about the sorting of cargo into ILVs and variation in the quantity and size of ILVs. This indicates that exosome biogenesis may be a coordinated process involving both ESCRT-dependent and -independent pathways [29]. 

## 3. Secretion and Uptake of Exosomes

The release of exosomes into the extracellular space depends on the transport of secretory MVBs and the fusion of the cell membrane after the inward budding of ILVs, which requires several key factors including the cytoskeleton (microtubules and microfilaments), molecular movements (mediated by kinetin and kinesin), molecular switches (small GTPase), and membrane fusion devices (soluble NSF attachment protein receptor [SNARE] complexes) [30]. During secretory MVB transport, the MVB moves along the microtubule cytoskeleton—a process that requires molecular motors for directed transport [31,32]. Microtubules and their associated molecular motors showed obvious polarity distribution in cells and were combined with the MVB. The MVB and plasma membrane were fused by Rab and its effector. Rab GTPase is an important factor, with more than 70 subtypes on the membrane surface involved in the regulation of vesicular functions such as budding, movement, and fusion [33]. Although the details of the fusion process remain elusive, the SNARE protein family was widely accepted as the core machinery for membrane fusion. This protein family includes vesicle SNARE (v-SNARE), which forms a complex with the homologous target SNARE (t-SNARE). Notably, this complex drives the fusion of two membranes in a zipped manner [34,35]. Through this process, MVBs fuse with the plasma membrane and release the exosomes into the extracellular space.

Signals from exosomes are generally transmitted to receptor cells through three different mechanisms: endocytosis, direct membrane fusion, or receptor–ligand interaction. Endocytosis is the primary method of exosome uptake and can be mediated by grid proteins, caveolae, or lipid rafts depending on the specific receptor cell type [36]. During endocytosis, exosomes may subsequently merge into endosomes or be transferred to lysosomes for degradation [37]. Additionally, the exosomal membrane can fuse directly with the plasma membrane of the receptor cell and release its contents, or bind to homologous receptors on the receptor cell membrane to subsequently trigger a cascade of intracellular signal transduction reactions [38].

## 4. Exosomal RNAs

### 4.1. Exosomal mRNAs

Exosomal mRNAs are important regulators of cellular biological processes. Exosomal mRNAs were first identified in mouse MC/9 and human HMC-1 cell lines using microarray analysis. Interestingly, exosomal mRNAs from mouse mast cells could be transferred into human mast cell lines, indicating that exosomes are effective vessels for the delivery of mRNA to other cells. It was further discovered that mRNAs were selectively taken up into exosomes because 270 transcripts were only detected in exosomes other than the donor cells (MC/9). Additionally, exosomal mRNAs were translated into functional proteins in the recipient cells, suggesting that exosomal mRNAs retain their function in recipient cells. These results demonstrate that exosomal mRNAs are critical mediators of intercellular communications [2].

Exosomal mRNAs also have advantages as biomarkers. First, exosomal mRNA can reflect the conditions of donor cells and are easy to detect since some exosomes (e.g., blood exosomes) circulate throughout the entire body [39]. In addition, the membrane of the exosome can protect exosomal mRNA from digestion by RNases [40]. For example, urinary exosomal mRNAs can remain stable for as long as two weeks at 4 °C [41]. Finally, exosomal mRNAs can affect the function of recipient cells more directly than exosomal ncRNAs since they can be translated into proteins in recipient cells. 

Exosomal mRNAs are considered as a critical indicator of cancers. Previous studies reported that tumor cells can express tumor-specific mRNAs or change the expression levels of normal exosomal mRNAs. For example, in glioblastoma, epidermal growth factor receptor (*EGFR*) is expressed by the tumor-specific mRNA *EGFRvIII*, which was recommended as a diagnostic biomarker for glioblastoma [42]. As another example, telomerase is considered a hallmark of cancer [43]. Human telomerase reverse transcriptase (*hTERT*) is generally not expressed in healthy humans. However, *hTERT* is detectable in multiple cancers, such as acute myelocytic leukemia, Burkitt lymphoma, and chronic lymphocytic leukemia, indicating that serum exosomal *hTERT* mRNA may be a potential pan-cancer biomarker [44]. Additionally, serum exosomal heterogeneous nuclear ribonucleoprotein H1 (*hnRNPH1*) mRNA levels in hepatocellular carcinoma (HCC) patients were significantly higher than those in control groups. Thus, exosomal *hnRNPH1* was suggested as a potential biomarker for HCC diagnosis [45].

Exosomal mRNAs were also suggested as biomarkers for diagnosing other diseases, such as those related to the human central nervous system and urinary system. A study including 20 older healthy adult subjects (≥65 years) and 20 younger healthy adult subjects (21–45 years) indicated that amyloid-β1-42 peptide (Aβ)–the main component of the amyloid plaques found in the brains of patients with Alzheimer’s disease [46] stimulated the release of exosomal cytokine mRNAs via macrophages and CD4 memory T-cells, indicating that exosomal cytokine mRNAs could potentially act as diagnostic biomarkers for Alzheimer’s disease [47]. Furthermore, Lv et al. suggested the urinary exosomal mRNA CD2 associated protein (*CD2AP*) as a biomarker for the diagnosis of kidney disease since a decrease in its expression level reflects the severity of tubulointerstitial fibrosis and glomerulosclerosis [48].

Exosomal mRNAs can also be used as biomarkers for the evaluation of drug resistance., which is currently one of the major challenges in cancer therapy. Shao et al. analyzed the exosomal mRNAs in the serum of 32 individuals (17 glioblastoma multiforme patients and 15 healthy individuals) and found that the exosomal O-6-methylguanine-DNA methyltransferase (*MGMT*) and N-methylpurine DNA glycosylase (*APNG*) mRNA levels were correlated with the levels of temozolomide resistance and the treatment efficacy in glioblastoma multiforme patients [49].

Based on studies of intracellular mRNAs, two successful commercial kits use urinary exosomal mRNAs (SAM pointed domain-containing ETS transcription factor (*SPDEF*) and ETS transcription factor (*ERG*)) and plasma exosomal mRNA (echinoderm microtubule-associated protein-like 4-anaplastic lymphoma kinase (*EML4-ALK*) fusion transcripts) to detect prostate cancer and nonsmall-cell lung cancer, respectively [50,51,52], which demonstrates the functionality of exosomal mRNAs as biomarkers. 

Some examples of exosomal mRNAs with the potential to be used as biomarkers for disease diagnosis are summarized in Table 1.

### 4.2. Exosomal miRNAs

MiRNAs, a class of small noncoding RNAs with a length of ~22 nt, play a principal role in the regulation of gene expression at the post-transcriptional level [61]. MiRNAs mainly function by binding to the 3′untranslated region (3′-UTR) of target mRNAs and inducing cleavage or reducing translation [62]. Both cellular miRNAs and exosomal miRNAs are involved in various biological activities, including cancer progression, immune responses, and cell cycle progression [61].

Exosomal miRNAs were proposed as potential biomarkers for diagnosing and predicting diseases. This primarily relies on their ability to reflect the internal conditions of cells, including physiological and pathological conditions [63]. In addition, miRNAs are the most abundant RNA molecules in exosomes, which makes their detection easier than that of other types of exosomal RNA [64]. Lastly, exosomal miRNAs show improved stability due to the protection afforded by their encapsulating membranes. It was observed that exosomal miRNAs can remain stable for five years when stored at −20 °C or for 14 days at 4 °C, with this stability being unaffected by repeated freezing and thawing [65]. These characteristics help increase the sensitivity of exosomal miRNA-based biomarkers. This is critical as limited sensitivity results in low detectability.

Recent studies identified some exosomal miRNAs with great potential for diagnosing cancers. Exosomes derived from cancer cells might affect the function of normal cells via miRNAs and could be an essential factor driving cancer metastasis. In a study on brain cancer, human and mouse tumor cells were observed to stop expressing phosphatase and tensin homolog (*PTEN*)—an important tumor suppressor—after dissemination to the brain due to inhibition mediated by exosomal *miR-19* that was secreted by astrocytes, which indicates that exosomal *miR-19* might be a suitable biomarker for diagnosing brain cancer metastasis [66]. Another study showed that colorectal cancer cells promoted the M2 pole of macrophages by transferring a set of miRNAs (*miR-25-3p*, *miR-130b-3p*, and *miR-425-5p*) through exosomes in response to stromal cell-derived factor 1/C-X-C chemokine receptor type 4 (*CXCL12/CXCR4*) activation through the *PTEN/PI3K/Akt* pathway, which enhanced the liver metastasis of colorectal cancer in vitro and in vivo [67].

Exosomal miRNAs can serve as biomarkers for diagnosing diseases other than cancer. Macrophages are critical for the maintenance of metabolic homeostasis, and their exosomal miRNAs are closely related to various metabolic-related diseases, such as diabetes and obesity [68]. For instance, exosomal *miR-690* binds to the 3′-UTR of the NAD kinase (*NADK*) mRNA, which is responsible for regulating insulin signaling and macrophage inflammation, to enhance insulin sensitivity [69]. Exosomal miRNAs could also reflect the viral infection of cells. One well-known example is the Epstein–Barr virus (EBV), the first human virus that was found to encode miRNAs [70]. B cells infected with EBV secrete exosomes containing EBV-miRNAs, which affect gene expression in the recipient cells [71]. 

Other recent examples of exosomal miRNAs with the potential to be used as biomarkers for disease diagnosis are summarized in Table 2. 

### 4.3. Exosomal lncRNAs

Long noncoding RNAs (lncRNAs) are a type of noncoding RNA longer than 200 nt [99]. LncRNAs are involved in the regulation of gene expression in diverse manners at multiple levels, such as gene transcription control, chromatin structure modulation, RNA splicing regulation, miRNA sponging, and RNA-binding protein interaction [100]. 

Typically, exosomal lncRNAs display strong tissue specificity and poor conservation, and the expression levels of exosomal lncRNAs can indicate the health conditions affecting tissues and cells, making them suitable for use as biomarkers [101,102]. Additionally, the large number of tissue- and cell-specific lncRNAs provides many options for diagnostic biomarkers. 

Exosomal lncRNAs were suggested as biomarkers to diagnose diseases. For example, lncRNA prostate cancer antigen 3 (lnc*PCA3*)—found in urinary exosomes—was approved as a biomarker to diagnose human prostate cancer by the US Food and Drug Administration. Exosomal lnc*PCA3* shows a much higher expression level in prostate cancer cells than in inflamed or normal prostate tissue (up to 70- to 100-fold), making lnc*PCA3* an efficient biomarker for diagnosing prostate cancer [103,104,105]. Another example is H19, a well-known oncogenic lncRNA found in serum exosomes that was significantly upregulated in bladder cancer patients when compared to that of healthy individuals, which highlights its potential use as a biomarker for bladder cancer [106,107].

Compared to exosomal miRNA, the study of exosomal lncRNA is in its infancy. According to GENCODE, there are more than 16,000 lncRNA genes in the human genome, which are estimated to produce more than 10,000 lncRNA transcripts, indicating a considerable candidate pool of potential diagnostic biomarkers [108,109]. 

Recent examples of exosomal lncRNAs that are potential diagnostic biomarkers are summarized in Table 3.

### 4.4. Exosomal circRNAs

Circular RNAs (circRNAs) are a subset of noncoding RNAs that lack 5′ caps and 3′ poly(A) tails and instead have a closed-loop structure [126]. As a class of endogenous RNAs, circRNAs are involved in various biological processes, including alternative splicing, transcription regulation, miRNA sponging, protein scaffolding, interacting with RNA-binding protein (RBP), and pseudogene creation [127,128,129,130,131]. Due to their multiple functions, circRNAs were closely linked to many diseases, such as cancers, neurodegeneration, diabetes, cerebrovascular diseases, and cardiovascular diseases. Thus, the expression of circRNAs could reflect the presence of these diseases [132,133,134,135]. Additionally, the closed-loop structure provides circRNAs with resistance to exoribonucleases and a long half-life [136,137,138]. These characteristics make exosomal circRNAs ideal biomarkers for disease diagnosis.

Exosomal circRNAs were studied as biomarkers for cancer diagnosis. RNA-seq analysis demonstrated that exosomal circRNAs enter circulation and are enriched at least two-fold in exosomes when compared to their levels in donor cells [139]. A study of exosomal circRNAs in colorectal cancer patients and healthy individuals revealed 67 absent circRNAs and 257 new circRNAs in patient serum exosomes, indicating the potential of exosomal circRNAs to act as biomarkers for the diagnosis of colorectal cancers [139]. Another study using microarray sequencing found a significant decrease in plasma exosomal *circ-0051443* in patients with hepatocellular carcinoma (HCC), and it was shown that *circ-00551443* releaseed *BCL2* antagonist/killer 1 *(BAK1*), which initiated cell apoptosis to prevent HCC via sponging *miR-331-3p* [140]. As a result, *circ-0051443* was considered a tumor suppressor and a novel potential biomarker for HCC diagnosis [140]. As another example, plasma exosomal *circ-133* can be used as a biomarker to monitor colorectal tumor progression, as exosomal *circ-133* expression induced by hypoxia was able to sponge *miR-133a* to activate the *GEF-H1/RhoA* axis in normoxic colorectal cancer cells, leading to the migration of colorectal cancer cells [141].

Exosomal circRNAs can also act as diagnostic biomarkers for nervous system diseases, ischemic diseases, and cardiovascular diseases. In cerebrospinal fluid, 26 exosomal circRNAs were shown to have significantly different expression levels in patients with immune-mediated demyelinating disease (IMDD) when compared to that of healthy controls, while the upregulations of *hsa_circ_0087862* and *hsa_circ_0012077* were recommended as potential diagnostic biomarkers for IMDD [138]. After ischemia, vascular smooth muscle cells secrete exosomal circRNA *cZFP609*, which is delivered into endothelial cells, resulting in reduced vascular endothelial growth factor A (*VEGFA*) expression and disrupted endothelial angiogenic function via the interaction with and sequestration of hypoxia-inducible factor 1 subunit alpha (HIF1α) [142]. In this case, *cZFP609* may act as a suitable biomarker to assess the clinical outcome and prognosis of ischemic diseases [142]. 

As a more recently discovered type of exosomal ncRNA, the study of exosomal circRNAs is in an early phase. Although research on exosomal circRNAs is mainly focused on cancer, it was reported that circRNAs are involved in nearly all aspects of biological activities. More exosomal circRNAs will likely be used in the future as biomarkers for the diagnosis of diseases or physiological/pathological processes. 

Some examples of exosomal circRNAs with the potential to be used as disease biomarkers are summarized in Table 4.

## 5. Exosome Isolation Techniques

Critical steps in the functional investigation of exosomes include their enrichment and isolation, which are necessary due to the small size and low density of exosomes as well as the heterogeneity of the bodily fluids in which they are found. To date, six main strategies exist to isolate exosomes from a diverse range of cellular detritus and interfering components, including ultra-speed centrifugation (differential ultracentrifugation and density-gradient ultracentrifugation), immunoaffinity capture, ultrafiltration, size-exclusion chromatography, polymer precipitation, and microfluidics-based techniques [154].

### 5.1. Ultra-Speed Centrifugation 

#### 5.1.1. Differential Ultracentrifugation

Differential ultracentrifugation is the most common method used to isolate exosomes. The principle of differential centrifugation is that various extracellular components with different densities, sizes, and shapes have different sedimentation rates under centrifugal force. Samples are typically centrifuged at a low speed (e.g., 300× *g*) to eliminate dead cells and debris [155,156]. Then, the supernatant is centrifuged at 2000× *g*, 10,000× *g*, and 100,000× *g* to pellet cell debris, apoptotic bodies, and protein aggregates, respectively [154]. Differential ultracentrifugation can isolate a large amount of material at a low cost. Due to this advantage, centrifugation was widely used to isolate exosomes from various biofluids, including cell culture medium, plasma, serum, saliva, and urine [157,158,159]. However, differential ultracentrifugation requires expensive equipment and a large amount of time. Additionally, this method may result in the coprecipitation of exosomes with other particles [160]. 

#### 5.1.2. Density-Gradient Ultracentrifugation

Density-gradient centrifugation separates contents based on their buoyant density in different solutions, such as sucrose and iodixanol [161,162]. A commonly used protocol for density-gradient centrifugation begins with loading 4 mL of tris/sucrose/D_2_O solution into the bottom of an SW 28 tube, carefully adding 25 mL of PBS containing partially isolated exosomes to the top of the sucrose cushion, and then centrifuging for 75 min at 100,000× *g* at 4 °C [163]. Thereafter, approximately 3.5 mL of the Tris/sucrose/D_2_O cushion is removed from the centrifuge tube and transferred to a fresh centrifuge tube [163]. Then, the mixture is diluted with 60 mL of PBS and centrifuged for 70 min at 100,000× *g* at 4 °C [163]. The pellet obtained as a result of this procedure includes the separated exosomes, which should be resuspended in 50–100 mL of PBS [163].

The advantages of this method are the high purity of the resulting products and the ability to separate subpopulations of exosomes. However, density-gradient centrifugation requires expensive ultracentrifugation equipment, which also causes sample loss during isolation. Additionally, density-gradient centrifugation could take as long as two days to isolate different products, which is not time-efficient in comparison to that of other isolation methods [164,165].

### 5.2. Immunoaffinity Capture

Immunoaffinity capture is based on the binding specificity between proteins and their corresponding antibodies. Typically, exosome marker proteins, especially transmembrane proteins, including CD9, CD8, CD63, and Rab5, are used to isolate exosomes [166,167]. In one protocol, the exosome pellet is passed through a column containing beads coated with antibodies against CD63, CD9, and CD8 [168]. Then, the antibody beads are washed to separate different exosomal populations. This approach is mostly employed for the further separation of exosomes after they were isolated using a centrifugation process. This approach allows the separation of distinct exosomal populations based on the presence or absence of certain protein markers. On the other hand, this method may lead to the loss of exosomes that lack specific protein markers. Additionally, the number of antibody-coated beads required for exosome separation by immunoprecipitation is proportional to the sample volume employed, which may be expensive.

### 5.3. Ultrafiltration

Ultrafiltration is a technique that employs porous membranes to capture molecules or particles of a given size, where smaller molecules and particles are allowed to pass through a membranous filter, while larger molecules and particles are trapped [169]. In one approach for exosome isolation, larger particles were initially removed by employing filters with pore widths of 0.8 and 0.45 microns, resulting in a filtrate with a high concentration of exosomes. Thereafter, smaller vesicles were removed from the filtrate by passing it through membranes with holes that were smaller than the desired exosomes (0.22 and 0.1 µm), and the eluate was discarded as waste. The size range of the exosomes acquired from the various pore filtration steps was characterized by the maximum and minimum sizes of the exosomes [170]. This approach can be employed as a supplement to ultracentrifugation to separate large microvesicles and exosomes; however, it can also be utilized as a standalone method. 

Exosome isolation may also be accomplished using cross-flow filtration (also known as tangential-flow filtration [171]), which is a technique based on consecutive filtrations and the use of nano-ultrafiltration. This process begins with a dead-end filtration of the cells and their detritus, followed by the filtration of large vesicles with a diameter of 1000 nm. Tangential flow-based filtering is used to eliminate impurities (mainly proteins) with a diameter smaller than the size cutoff, which are then disposed of in a trash chamber. The filtrate, which contains exosomes, is then passed through the exclusion filter many times, resulting in a concentrated input solution. Finally, exosomes are further separated using a track-etched membrane with a diameter ranging from 50 to 250 nm and a defined and uniform pore size track [170].

Ultrafiltration can isolate samples with high efficiency beyond the volume limitation, which makes this method a useful substitute for ultracentrifugation [172]. Compared to that of ultracentrifugation, ultrafiltration requires significantly less time. However, there is a risk of filter clogging occurring during analysis, which reduces the lifetime of the membranes [173]. Additionally, the shear forces involved in the process could disrupt the integrity of exosomal membranes, leading to exosome lysis and particle deformation [174].

### 5.4. Size-Exclusion Chromatography

Size-exclusion chromatography is used to separate exosomes from other extracellular vesicles according to their size, and it is conducted using the same method employed for protein separation [175]. The column used for size-exclusion chromatography is filled with a porous stationary phase through which tiny particles may pass. This penetration is responsible for the slower flow of the smaller particles down the tube, which causes them to elute later in the gradient and after the larger particles [176,177].

Size-exclusion chromatography can be used to purify exosomes while preserving the vesicle structure, integrity, and biological activity. The isolated exosomes also maintain their proper vesicle characteristics due to gravity manipulation, which avoids the damage caused by shear forces. Although this increased quality comes at the expense of the overall yield of exosomes, size-exclusion chromatography procedures can be scaled up to obtain higher yields. A substantial amount of initial biofluid is necessary to compensate for the lower yield if the recovery rate is only moderate [178]. In addition, size-exclusion chromatography could reduce the production of exosomal mRNA and exosomal protein. 

### 5.5. Polymer Precipitation

The presence of highly hydrophilic polymers can create a hydrophobic micro-environment through their interaction with the water molecules surrounding exosomes, leading to exosome precipitation [179]. Polyethylene glycol (PEG) with weights ranging from 6000 to 20,000 Da is widely used for the polymer-based exosome precipitation method. In this method, large contaminant particles, including debris and apoptotic bodies, are first removed during a pretreatment step [180]. The samples are then incubated with a PEG solution at 4 °C overnight to induce exosome precipitation [180]. Low-speed centrifugation (1500× *g*) is then used to collect the precipitated exosomes [180].

Lectin precipitation is an alternative to PEG precipitation. Lectins bind the carbohydrate moieties of other particles with extreme specificity. Lectins are thus able to attach to carbohydrates on the surface of exosomes and affect their solubility, making the exosomes insoluble and causing them to precipitate out of solution [181].

Precipitation using highly hydrophilic polymers is rapid, simple, low-cost, and does not require complicated equipment. However, the final exosome pellet becomes contaminated, which prevents further omics-based analysis for exosomes. Although this makes the precipitation method ineffective in clinical research settings, it is effective in other situations.

### 5.6. Microfluidics-Based Techniques

Tremendous recent developments in microfabrication technology enabled the creation of lab-on-a-chip-type microfluidic devices for effective exosome separation [182,183,184]. These small microfluidic machines enable exosome separation from fingertip amounts of bodily fluids and exosome characterization for in situ diagnoses. Microfluidic technology is revolutionizing exosome-based diagnostics by combining what is usually a two-step approach involving exosome separation and characterization into one step [184]. The immuno-microfluidic approach is a widely used microfluidics-based method that is comparable to the immunoaffinity capture isolation method. Exosomes are isolated via the specific binding between protein markers and the microfluidic devices. For example, the ExoChip is a popular microfluidic device that was employed in conjunction with CD63 antibodies [185].

The benefits of microfluidic technology include efficient and rapid processing and the high purity of resulting exosomes. More importantly, microfluidic techniques can isolate exosomes based on their physical and biochemical properties simultaneously. Apart from the requirement for specialized equipment, this method suffers from many of the same disadvantages as the aforementioned immunoaffinity capture method described previously. Although microfluidics-based techniques were not accepted as a standardized exosome isolation method, they have great potential for use in the future. 

Here, the advantages and disadvantages of above exosome isolation techniques are summarized in Table 5.

## 6. Conclusions

Exosomal mRNAs, miRNAs, lncRNAs, and circRNAs are important mediators of intercellular communication, and they will absolutely provide essential clues and huge opportunities in disease diagnosis in the near future. The investigation of exosomal biomarkers highlighted their great value in diagnosis and prognosis since they avoid the limitations of conventional solid biopsy, especially by reducing the trauma associated with surgery. 

However, the exploration of exosomal biomarkers is still in the early phase, and a ubiquitous clinical application is greatly limited.

Although the convenience of exosomal biomarkers is undisputed, challenges still remain with biomarker selection, and no biomarker can achieve 100% accuracy. Future efforts should focus on identifying the most significant changed biomarkers to improve the accuracy of disease diagnosis. Therefore, detailed mechanisms of exosomal RNAs require further investigation.

To analyze clinical samples on a large scale in the feature, it is necessary to extract exosomes more rapidly, accurately, and completely, thereby implementing the use of exosomes as new biomarkers into clinical practice. To date, there is no uniform standard for the isolation, purification, or quality assessment of exosomes, which also restricts the progress of research on exosomal diagnostic technology. Therefore, further study is required to develop new methods to enhance the exosome isolation and purity, which would greatly benefit the research on the clinical applications of exosomes as disease biomarkers. Despite the insights highlighted in this review, a more comprehensive view of exosomal biomarkers requires further investigation. 

Challenges still remain in the field of exosomal RNAs as biomarkers. However, exosomal RNAs will definitely provide exciting new insights and outline promising fields for the development of novel therapeutic strategies.

## Figures and Tables

**Figure 1 ijms-23-02461-f001:**
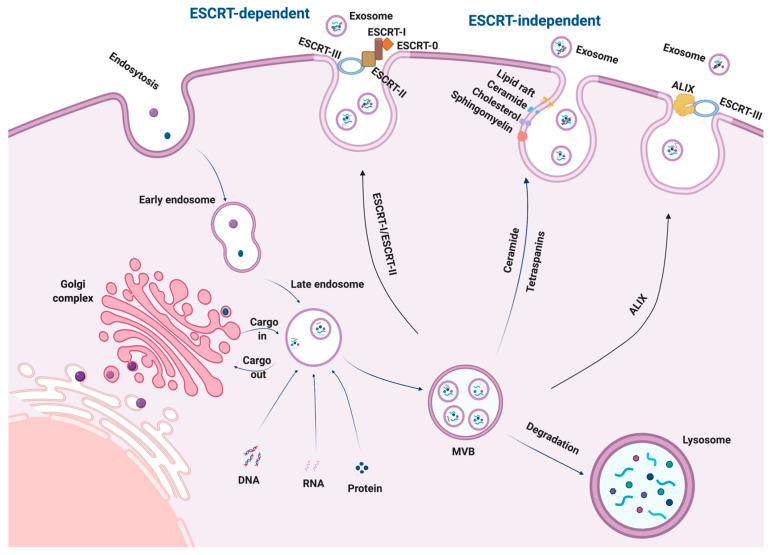
Biogenesis of exosomes.

**Table 1 ijms-23-02461-t001:** Summary of exosomal mRNAs as potential disease biomarkers.

Exosome Sources	Diseases	Potential Biomarkers (mRNA)	References
Serum & glioblastoma CCM	Glioblastoma	*EGFRvIII*	[42]
Urine	Tubulointerstitial fibrosis & glomerular sclerosis	*CD2AP*	[48]
Serum & GBM CCM	Temozolomide resistance in GBM	*MGMT* & *APNG*	[49]
Urine	Prostate cancer	*ERG*, and *SPDEF*	[50]
U87 & A172 CCM	Temozolomide chemoresistance in glioblastoma	*PTPRZ1-MET*	[53]
Serum	Hepatocellular carcinoma	*hnRNPH1*	[45]
Serum	Docetaxel resistance in prostate cancer	*CD44v8-10*	[54]
HFF CCM	Toxoplasma-infected HFFs	*RAB-13*, *EEF1A1*, *TMSB4X* & *LLPH*	[55]
Serum	Colorectal cancer	*KRAS* mutation & *BRAF* mutation	[56]
Serum	Gastric cancer	*MT1-MMP*	[57]
Plasma	Resistance to hormonal therapy in prostate cancer	*AR-V7*	[58]
Serum	Pancreatic ductal adenocarcinoma	*WASF2*, *ARF6*, *SNORA74A* & *SNORA25*	[59]
Serum & CCM	Acute lymphoblastic leukemia	*DNMT1*	[60]

CCM: cell culture media; GBM: human glioblastoma multiforme; HFF: human foreskin fibroblasts; EGFRvIII: epidermal growth factor receptor variant III, CD2AP: CD2 associated protein; MGMT: O-6-methylguanine-DNA methyltransferase; APNG: N-methylpurine DNA glycosylase; ERG: ETS transcription factor; SPDEF: SAM pointed domain containing ETS transcription factor, PTPRZ1: protein tyrosine phosphatase receptor type Z1; MET: MET proto-oncogene, receptor tyrosine kinase; hnRNPH1: heterogeneous nuclear ribonucleoprotein H1; CD44v8-10: isoform of cluster of differentiation 44 variant, and contains the variant exons 13–15 (v8–v10); RAB-13: RAB13, member RAS oncogene family; EEF1A1: eukaryotic translation elongation factor 1 alpha 1, TMSB4X: thymosin beta 4 X-linked; LLPH: LLP homolog, long-term synaptic facilitation factor, KRAS: KRAS proto-oncogene, GTPase; BRAF: B-raf proto-oncogene, serine/threonine kinase; MT1-MMP: mmbrane type-1 matrix metalloproteinase; AR-V7: androgen receptor variant 7; WASF2: WASP family member 2; ARF6: ADP ribosylation factor 6; SNORA74A: small nucleolar RNA, H/ACA box 74A; SNORA25: small nucleolar RNA, H/ACA box 25; DNMT1: DNA-methyltransferase 1.

**Table 2 ijms-23-02461-t002:** Summary of exosomal miRNAs as potential disease biomarkers.

Exosome Sources	Potential Biomarkers	Diseases	Target Genes/Pathways	Effects	References
Serum	*miR-193b*	AD	*APP*	Inhibits AD development	[72]
Glioblastoma stem CCM	*miR-9*	Antiangiogenic therapy for glioblastoma	*RGS5*, *SOX7* & *ABCB1*	Promotes angiogenesis	[73]
Plasma	*miR-146a*	Heart failure	*IRAK-1*, *TRAF6*, *NOX-4 SMAD4* & *TGF-β*	Promotes the proliferation and inhibit the apoptosis of cardiomyocytes	[74]
Plasma	*miR-21* & *miR-181a-5p*	Thyroid cancer	N/A	Distinguishes between follicular and papillary thyroid cancer	[75]
HCT116 CCM & serum	*miR-25*, *miR-130b*, *and miR-425*	Colorectal cancer	*PTEN/PI3K/AKT* pathway	Promotes the liver metastasis of colorectal cancer	[67]
CCM & serum	*miR-1247-3p*	Liver cancer	*B4GALT3*	Promotes the lung metastasis of liver cancer	[76]
A2780 CCM	*miR-223*	Epithelial ovarian cancer	*PTEN**/**PI3K/AKT* pathway	Promotes chemoresistance	[77]
Multiple sources	*miR-21*	Various cancers	Multiple targets	Promotes cancer development	[78,79,80,81,82]
Microglia culture media	*miRNA-137*	Ischemic brain injury	*NOTCH1*	Promotes neuroprotection	[83]
Plasma	*miR-125a-5p/miR-141-5p*	Prostate cancer	N/A	N/A	[84]
Serum	*miR-7977*	Lung adenocarcinoma	N/A	Promotes proliferation and invasion, and inhibits apoptosis of A549 cells	[85]
Pan02 CCM	*miR-155-5p* & *miR-221-5p*	PDAC	*E2F2*	Promotes PDAC progression	[86]
Cardiac telocyte CCM	*miR-21-5p*	Myocardial infarction	*CDIP1*	Promotes angiogenesis	[87]
HT-29/SW480 CCM	*miR-375-3p*	Colon cancer	N/A	Regulates EMT of colon cancer cells	[88]
MSC CCM	*miR-542-3p*	Cerebral infarction	*TLR4*	Inhibits inflammation and cerebral infarction	[89]
CCa CCM & serum	*miR-1468-5p*	Cervical cancer	*HMBOX1* & *JAK2/STAT3* pathway	Promotes tumor immune escape	[90]
MSC CCM	*miR-21-5p*	Breast cancer	*S100A6*	Promotes chemoresistance	[91]
Plasma	*miR-1-3p*	Sepsis	*SERP1*	Induces endothelial cell dysfunction	[92]
Plasma	*miR-451a* & *miR-21-5p*	AD	N/A	N/A	[93]
hUCMSC CCM & serum	*miR-139-5p*	Bladder cancer	*PRC1*	Inhibits tumorigenesis	[94]
OSCC CCM & blood	*miR-340-5p*	OSCC	*KLF10*	Promotes radioresistance	[95]
Saliva	*miR-24-3p*	OSCC	*PER1*	Maintains the proliferation of OSCC cells	[96]
Saliva	*miR-134* & *miR-200a*	OSCC	N/A	N/A	[97]
Serum	*miR-1226*	PDAC	N/A	N/A	[98]

*APP*: amyloid precursor protein; *RGS5*: regulator of G protein signaling 5; *SOX7*: SRY-box transcription factor 7; *ABCB1*: ATP binding cassette subfamily B member 1; *SMAD4*: *SMAD* family member 4; *TGF-β*: transforming growth factor beta 1; *B4GALT3*: beta-1;4-galactosyltransferase 3; *PTEN*: phosphatase and tensin homolog; *NOTCH1*: Notch Receptor 1; *E2F2*: *E2F* transcription factor 2; *CDIP1*: cell death inducing P53 target 1; *TLR4*: toll-like receptor 4; *HMBOX1*: homeobox containing 1; *JAK2*: janus kinase 2; *STAT3L*: signal transducer and activator of transcription; 3S100A6: S100 calcium binding protein A6; CCM: cell culture media; SERP1: stress associated endoplasmic reticulum protein 1; EBV: epstein-barr virus; AD: alzheimer’s disease; *PRC1*: polycomb repressor complex 1; *KLF10*: kruppel like factor 10; *PER1*: period circadian regulator 1; *HCT116*: human colorectal carcinoma reporter gene cell lines; A2780: human epithelial ovarian cancer cell line A2780; hUCMSCs: human umbilical cord mesenchymal stem cells; A549: human LUAD cell line; PDAC: pancreatic ductal adenocarcinoma; EMT: epithelial–mesenchymal transition; MSC: mesenchymal stem cell; CCa: cholangiocarcinoma; OSCC: esophageal squamous cell carcinoma.

**Table 3 ijms-23-02461-t003:** Summary of exosomal lncRNAs as potential disease biomarkers.

Exosome Sources	Potential Biomarkers	Diseases	Effects	Mechanistic Approaches	References
Plasma	*Linc-POU3F3*	PD	N/A	N/A	[110]
Plasma	*lnc-MKRN2-42:1*	PD	Affects the occurrence and development of PD	N/A	[111]
Various PC CCM & serum	*lncRNA-UCA1*	PC	Promotes angiogenesis	*miR-96-5p*/*AMOTL2* axis	[112]
Plasma	*BACE1-AS*	AD	N/A	N/A	[113]
Serum	*HOXD-AS1*	Prostate cancer	Promotes metastasis	*miR-361-5p/FOXM1* axis	[114]
Serum	*SNHG16*	Breast cancer	Inhibits immunity	*miR-16–5p/SMAD5* axis	[115]
Serum	*lncUFC1*	NSCLC	Promotes proliferation, migration, and invasion	Inhibits *PTEN* expression via *EZH2*-mediated epigenetic silencing	[116]
Urine	*lncBCYRN1*	Bladder cancer	Promotes lymphatic metastasis	Activates *WNT5A/VEGF-C/VEGFR3* feedforward loop	[117]
Urine	*lncLNMAT2*	Bladder cancer	Promotes lymphatic metastasis	N/A	[118]
Primary MSCs CCM	*LINC01559*	GC	Promotes progression	Multiple approaches	[119]
GC CCM & serum	*lncRNA-GC1*	GC	N/A	N/A	[120]
GC CCM	*lncPCGEM1*	GC	Promotes invasion and metastasis	Maintains the stability of *SNAI1*	[121]
Urine	*TERC*	BLCA	N/A	N/A	[122]
M1/M2 macrophage CCM	*lncAFAP1-AS1*	Esophageal cancer	Promotes migration and metastasis	miR-26a/*ATF2* axis	[123]
MSCs CCM	*MALAT1*	DICS	Promotes mitochondrial metabolism and rejuvenation	*miR-92a-3p*/*ATG4a* axis	[124]
Serum	*H19*	Breast cancer	Reduce DOX resistance	N/A	[125]

PD: Parkinson’s disease; CCM: cell culture media; PC: pancreatic cancer; AD: alzheimer’s disease; NSCLC: non-small-cell lung cancer; GC: gastric cancer; AMOTL2: angiomotin like 2; FOXM1: forkhead box M1; SMAD5: SMAD family member 5; EZH2: enhancer of zeste 2 polycomb repressive complex 2 subunit; WNT5A: wnt family member 5A; VEGF-C: vascular endothelial growth factor C; VEGFR3: vascular endothelial growth factor receptor 3; SNAI1: snail family transcriptional repressor 1; BLCA: Bladder urothelial car-cinoma, ATF2: activating transcription factor 2; TERC: telomerase RNA component; MALAT1: metastasis associated lung adenocarcinoma transcript 1; DICS: doxorubicin-induced cardiac senescence; ATG4a: autophagy related 4A cysteine peptidase; DOX: doxorubicin.

**Table 4 ijms-23-02461-t004:** Summary of exosomal circRNAs as potential disease biomarkers.

Exosome Sources	Potential Biomarkers	Diseases	Effects	Mechanistic Approaches	References
Serum	*circ-G042080*	Myeloma-related myocardial damage	Promotes autophagy	*miR-4268*/*TLR4* axis	[143]
Serum	*circGlis3*	Type 2 diabetes	Regulates islet EC function	Regulates GMEB1 degradation & *HSP27* phosphorylation	[144]
Plasma	*circ-RanGAP1*	Gastric cancer	Promotes metastasis and development	*miR-877–3p*	[145]
HCC CCM	*circRNA-100338*	HCC	Promotes angiogenesis and invasion	N/A	[146]
Plasma	*circRNA_0056616*	Lymph node metastasis in lung adenocarcinoma	N/A	N/A	[147]
Serum	*circ_0006156*	Thyroid cancer	Promotes tumorigenesis	*miR-1178*/*TLR4* axis	[148]
Serum	*circ_0075828, circ_0003828* & *circ_0002976*	HGA	N/A	N/A	[149]
Serum	*circRNA_104484* & *circRNA_104670*	Sepsis	N/A	N/A	[150]
Serum	*circ-ATP10A*	Multiple myeloma	Promotes angiogenesis	Multiple axises	[151]
K562 & K562/G01 CCM	*circ_0058493*	CML	Drug resistance	*miR-548b-3p*	[152]
GBM CCM	*circNEIL3*	Glioma	Promotes progression	Stabilizing *IGF2BP3*	[153]

CCM: cell culture media; islet EC: islet endothelial cells; HCC: hepatocellular carcinoma; HGA: high-grade astrocytoma; PC: pancreatic cancer; AD: Alzheimer’s disease; GC: gastric cancer; BMSCs: bone marrow-derived mesenchymal stromal cells; CML: chronic myeloid leukemia; TLR4: toll like receptor 4; GMEB1: glucocorticoid modulatory element-binding protein 1; HSP27: heat shock protein 27; GBM: glioblastoma multiforme; NEIL3: nei like DNA glycosylase 3; IGF2BP3: insulin-like growth factor 2 mRNA binding protein 3.

**Table 5 ijms-23-02461-t005:** Summary of advantages and disadvantages of exosome isolation techniques.

Isolation Techniques	Advantages	Disadvantages
Differential ultracentrifugation	Low cost	Requires expensive equipment
		Coprecipitation with other particles
	Suitable to isolate a large amount of material	Potential mechanical damage
	Low risk of reagent pollution;	Not appropriate for small volume diagnosis
Density-gradient ultracentrifugation	High purity of resulting products	Heavy workload, low recovery.
	Separating subpopulations of exosomes	Complicated steps
		Time-consuming
		Low recovery
Immunoaffinity capture	High specificity	Low extraction efficiency
		High cost of antibodies
	Simple operation	Potential pollution of pH and salt concentration
		Isolation for antibody-bound exosomes only
	No chemical pollution	Low processing volume
Ultrafiltration	Fast	Moderate exosome yield and purity
		Potential shear stress induced deterioration
	Cheap equipment cost	Possible exosome loss because of membrane trapping and clogging
Size-exclusion chromatography	High purity of resulting products	Moderate exosome yield
		High cost
	Fast	Time-consuming
		Require high-quality chromatographic column
		Require extra exosome enrichment step
Polymer precipitation	Easy to use	Low exosome purity
	No special equipment requirement	
	Multiple sample processing	Limiting further omics-based analysis
	Low cost	Require extra complicated clean-up steps
	Low risk of exosome damage	
Microfluidics-based techniques	Sample volume requirement	Low sample capacity
	Fast	
	Relative low cost	
	High detection sensitivity	
	Multifunctional operations integration	

## Data Availability

Not applicable.

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
