# Peer review of "Exosomal RNAs: Novel Potential Biomarkers for Diseases—A Review"

_ijms, 2022, doi:10.3390/ijms23052461_

Round 1

Reviewer 1 Report

Exosomes play important role in various human disease’s pathogenesis and participate in intercellular communication and may transfer various biomolecules between cells and tissues maintaining tissue homeostasis. Exosomes take part in various physiological and pathological conditions and may be used for targeted delivery of drugs and other biologically active substances. The authors discuss current progress and various technological advancements in exosome paying special attention to disease-associated RNAs biomarkers. This is an important area of biomedical investigation and the data of the manuscript will be interesting for the readers of IJMS. Some minor spelling and grammar checks needs to be done.

  1. Author should provide a table of different exosome isolation methods with their limitations.
  2. Conclusion is too general and does not contain any suggestions and future prospective.
  3. More up to date scientific literature should be included in the manuscript related to the field as is limited and this study will cover diverse scientific community.

Author Response

Response to Reviewer 1 Comments

Point 1: Author should provide a table of different exosome isolation methods with their limitations.

Response 1: Thank you for your valuable suggestion. We have provided a new table to show the advantages and disadvantages of different exosome isolation methods. We have added the table in the manuscript and attached it below.

Table 5. Summary of the advantages and disadvantages of exosome isolation techniques

Isolation Techniques

Advantages

Disadvantages

Differential ultracentrifugation

• Low cost

• Suitable to isolate a large amount of material

• Low risk of reagent pollution;

• Requires expensive equipment

• Co-precipitation with other particles

• Potential mechanical damage

• Not appropriate for small volume diagnosis

Density-gradient ultracentrifugation

• High purity of resulting products

• Separating subpopulations of exosomes

• Heavy workload, low recovery.

• Complicated steps

• Time-consuming

• Low recovery

Immunoaffinity capture

• High specificity

• Simple operation

• No chemical pollution

• Low extraction efficiency

• High cost of antibodies

• Potential pollution of pH and salt concentration

• Isolation for antibody-bound exosomes only

• Low processing volume

Ultrafiltration

• Fast

• Cheap equipment cost

• Moderate exosome yield and purity

• Potential shear stress induced deterioration

• Possible exosome loss because of membrane trapping and clogging

Size-exclusion chromatography

• High purity of resulting products

• Fast

• Moderate exosome yield

• High cost

• Time-consuming

• Require high-quality chromatographic column

• Require extra exosome enrichment step

Polymer precipitation

• Easy to use

• No special equipment requirement

• Multiple sample processing

• Low cost

• Low risk of exosome damage

• Low exosome purity

• Limiting further omics-based analysis

• Require extra complicated clean-up steps

Microfluidics-based techniques

• Sample volume requirement

• Fast

• Relative low cost

• High detection sensitivity

• Multi-functional operations integration

• Low sample capacity

Point 2: Conclusion is too general and does not contain any suggestions and future prospective.

Response 2: We appreciate the reviewer’s comments. We have revised the conclusion and added the suggestions and prospect.

Point 3: More up to date scientific literature should be included in the manuscript related to the field as is limited and this study will cover diverse scientific community.

Response 3: Thank you for your advice. We have added some latest literature about exosomes in the introduction section. Also, we have added some latest examples of exosomal mRNA, exosomal miRNA, exosomal lncRNA, and exosomal circRNA as biomarkers for diseases in Table 1, 2, 3, and 4, respectively.

Reviewer 2 Report

This is a comprehensive review the authors included all types of non-coding RNAs. We might have some similar reviews but here they have included many new findings which makes it different from other studies. 

comment: In tables 2, 3, and 4, authors may mention the mechanistic approaches each of these noncoding RNAs utilize to exert their function. For example, if promoting metastasis, how?   

Author Response

Point 1: In tables 2, 3, and 4, authors may mention the mechanistic approaches each of these noncoding RNAs utilize to exert their function. For example, if promoting metastasis, how?  .

Response 1: Thank you for your valuable suggestion. We have added the target genes of miRNA or related pathways in Table 2. Also, we added mechanitic approaches including target miRNAs and/or downstream axis in Table 3 and 4.